# Smart Agriculture and Greenhouse Gas Emission Mitigation: A 6G-IoT Perspective

**Sofia Polymeni** [1,*,†,‡], **Dimitrios N. Skoutas** [1,‡], **Panagiotis Sarigiannidis** [2,‡], **Georgios Kormentzas** [1,‡] **and Charalabos Skianis** [1,‡]

1 Department of Information and Telecommunication Systems Engineering, University of the Aegean, 83200 Samos, Greece; d.skoutas@aegean.gr (D.N.S.); gkorm@aegean.gr (G.K.); cskianis@aegean.gr (C.S.)

2 Department of Electrical and Computer Engineering, University of Western Macedonia, 50100 Kozani, Greece; psarigiannidis@uowm.gr

* Correspondence: spolymeni@aegean.gr

† Current address: Lymperis Building, 2 Palama & Gorgyras St., 83200 Samos, Greece.

‡ All authors contributed equally to this work.

**Abstract:** Smart farming has emerged as a promising approach to address the agriculture industry's significant contribution to greenhouse gas (GHG) emissions. However, the effectiveness of current smart farming practices in mitigating GHG emissions remains a matter of ongoing debate. This review paper provides an in-depth examination of the current state of GHG emissions in smart farming, highlighting the limitations of existing practices in reducing GHG emissions and introducing innovative strategies that leverage the advanced capabilities of 6G-enabled IoT (6G-IoT). By enabling precise resource management, facilitating emission source identification and mitigation, and enhancing advanced emission reduction techniques, 6G-IoT integration offers a transformative solution for managing GHG emissions in agriculture. However, while smart agriculture focuses on technological applications for immediate efficiency gains, it also serves as a crucial component of sustainable agriculture by providing the tools necessary for long-term environmental supervision and resource sustainability. As a result, this study also contributes to sustainable agriculture by providing insights and guiding future advancements in smart farming, particularly in the context of 6G-IoT, to develop more effective GHG mitigation strategies for smart farming applications, promoting a more sustainable agricultural future.

**Keywords:** 6G-enabled IoT (6G-IoT); greenhouse gas emissions (GHG); greenhouse gas mitigation; internet of things (IoT); smart agriculture

## 1. Introduction

The worldwide agricultural industry, while being essential for human livelihood and global economies, has also emerged as a significant contributor to greenhouse gas (GHG) emissions, underlining the urgency in finding sustainable farming methods, particularly in the realm of smart farming. However, although the integration of information and communication technologies into agricultural practices holds immense promise for enhancing sustainability and boosting production efficiency, there exists a notable gap in understanding the full impact of these technologies on GHG emissions [1]. This gap underscores the complexity of agriculture's role in the environmental field and the need for new and improved approaches in assessing the net impact of technological interventions.

The introduction of automated systems, sophisticated data analytics, and IoT-based environmental monitoring has undeniably improved agricultural productivity. However, this technological advancement has not consistently resulted in a proportional reduction in GHG emissions [2,3]. This necessitates a comprehensive evaluation of GHG emission metrics in smart farming, considering both direct emissions from traditional practices and indirect emissions from technology deployment and operation. GHG emission reduction is

a critical economic and social strategy that goes beyond environmental concerns. Agriculture's inherent reliance on natural resources and vulnerability to climatic fluctuations make emission reduction in this sector especially challenging. As a result, there is an ever-urgent need to develop and implement effective mitigation techniques that are both sustainable and resilient in the long term, covering a wide range of practices, from improved crop and soil management to the implementation of advanced technological solutions [2].

The integration of the Internet of Things (IoT) in agriculture represents a powerful tool for managing GHG emissions, particularly when combined with the capabilities of the newly introduced 6G wireless technology. At their core, IoT and 6G technologies exploit the use of interconnected devices and ultra-fast wireless networks to enable seamless data exchange and intelligent decision-making [4], offering new opportunities for optimizing resource management, enhancing productivity, and mitigating GHG emissions in agricultural applications. This synergy between IoT and 6G wireless communications, called 6G-enabled IoT (6G-IoT), promises high-definition and low-latency real-time monitoring, thus enabling multi-dimensional data analysis and facilitating more precise and informed decision-making [1]. Unlike 5G and its predecessors that laid the foundations for high-speed connectivity and the increase of IoT devices, this enhanced data capture and analysis allows for more sophisticated predictive analytics and machine learning (ML) algorithms, leading to the proactive implementation of more sustainable practices, thereby contributing to global efforts aimed towards GHG emission mitigation.

However, in the realm of agricultural innovation, two paradigms, "smart agriculture" and "sustainable agriculture", have emerged as critical pathways toward a more efficient, productive, and environmentally friendly future. Smart agriculture, as mentioned above, represents the application of cutting-edge technologies, including IoT and, more recently, 6G-IoT, to enhance the precision, automation, and decision-making processes in farming operations, leveraging real-time data acquisition, advanced analytics, and automation to optimize resource use and increase crop yields while minimizing environmental impacts [5,6]. On the other hand, sustainable agriculture is a broader concept that encompasses practices designed to preserve environmental health, maintain agricultural productivity over time, and minimize GHG emissions, among other goals. However, while smart agriculture focuses on technological applications for immediate efficiency gains, it also serves as a crucial component of sustainable agriculture by providing the tools necessary for long-term environmental supervision and resource sustainability [7,8].

This study expands upon our previous research on 6G-IoT integration in smart agriculture [9] by focusing on two key areas: identifying GHG indicators and exploring the potential of 6G-IoT to mitigate them. First, it delves into both livestock and crop production to understand the current GHG landscape and pinpoint areas where emissions have risen due to intensive practices and increased chemical reliance. Second, it connects this understanding to the emerging 6G-IoT paradigm, offering an overview of the developing technological landscape and concrete examples of how these technologies can be applied in practice to reduce GHG emissions within smart farming. Therefore, we anticipate that this work will encourage additional investigation and advancement, encouraging the agricultural industry to embrace technology for a more sustainable future.

The remainder of this paper is organized as follows. Section 2 provides a comprehensive overview of existing research on the topic, highlighting key findings and identifying gaps in knowledge to establish the context and scope of this review paper. Section 3 delves into the impact of GHG emissions in the agricultural sector, identifying key contributors commonly found in smart farming applications. Section 4 then presents commonly derived GHG emission mitigation metrics already used in farming practices. Subsequently, Section 5 highlights the potential of 6G-IoT technology for smart agriculture and its impact on reducing GHG emissions. Building on the foundation laid in the previous section, Section 6 delves deeper into specific ways these technologies can contribute to GHG mitigation strategies. Finally, Section 7 concludes the paper by summarizing important insights, discussing the interdisciplinary sustainability implications, and highlighting the

significance of 6G-IoT in promoting environmentally friendly smart farming practices. Furthermore, it provides the final conclusions of this study and outlines potential areas for future research.

## 2. Related Work and Survey Scope

The field of smart farming, integrating advanced technologies such as the IoT, stands at a crucial crossroad between agricultural productivity and environmental sustainability, especially regarding GHG emissions; while smart farming holds promise for boosting efficiency and productivity, its impact on GHG emissions remains a topic of ongoing research. To position this study within its broader research context, this section critically examines existing literature regarding various aspects of GHG emissions in agricultural practices. The studies discussed here offer insights into general approaches to mitigating GHG emissions in agriculture, spanning from climate-smart agricultural practices to detailed analyses of emissions from different farming systems. Consequently, this section provides a foundational understanding of the current knowledge landscape in the field, while also outlining the contributions of the current study.

Evidently, the need to reduce GHG emissions from agricultural practices has prompted numerous studies, each examining different aspects of this broad challenge. In this context, Qian et al. [10] highlighted the significant contribution of agriculture in GHG emissions, focusing on methane ($CH_4$) and nitrous oxide ($N_2O$), specifically for rice agriculture, while also providing valuable insights into post-harvest management and soil treatments as practical mitigation strategies. Similarly, Schwarz et al. [11] investigated various GHG emission mitigation strategies catered towards agricultural and horticultural systems, specifically focusing on the effects of extreme weather events and the assessment of nitrogen usage efficiency in dairy farms.

The work of Sejian et al. [12] also delved into GHG emission mitigation strategies, but with a solid focus on livestock, highlighting the need for comprehensive management strategies to address enteric fermentation and manure management. By emphasizing the substantial contributions of livestock to agricultural emissions, their research supports broader efforts for GHG emission mitigation. On that note, Nayak et al. [13] expanded this discussion by further examining the management opportunities to mitigate GHG emissions from Chinese agriculture, highlighting the potential in terms of both technology and financial gain across cropland, grassland, and livestock systems. Finally, the review paper of Llonch et al. [14] was also focused on GHG emission mitigation strategies in livestock; however, it also took animal welfare into account, being essential to understanding how GHG mitigation efforts in livestock management can align with or impact welfare standards. As a result, their work is crucial for developing balanced strategies that consider both environmental sustainability and ethical considerations in livestock production.

Alongside GHG mitigation efforts, the advancement of precision agriculture technologies offers a new approach to sustainability. In this context, Koutsos and Menexes [15] thoroughly examined the benefits of adopting such technologies, highlighting their agronomic, economic, and environmental benefits, by demonstrating the way that they can optimize returns on inputs while minimizing environmental impacts, thereby offering a path towards more sustainable farming practices. Furthermore, in advancing real-time decision-making in farm management, Roy and George K. [16] emphasized the importance of Geographic Information Systems (GISs), Global Positioning Systems (GPSs), as well as remote sensing, demonstrating the technology's potential to boost production and efficiency in agriculture. The work of Panchasara et al. [17] also focused on the application of precision agriculture practices to increase crop yield efficiency, emphasizing the significance of data-driven decision-making for maximizing agricultural results, while Cheng et al. [18] further explored the application of state-of-the-art technologies, including ML and artificial intelligence (AI), showcasing how these technologies support economical resource utilization and smaller environmental footprints.

## 2.1. GHG Emissions and Sustainable Agriculture

The pressing need to reduce GHG emissions has prompted the explorations of sustainable agricultural practices as a critical solution pathway. The relationship between GHG emissions and sustainable agriculture is multifaceted, encompassing both direct and indirect mechanisms through which agriculture impacts the global carbon cycle and climate change. Sustainable agriculture practices, including conservation tillage, mixed cropping systems with cover crops, and agroforestry, have been recognized for their potential to significantly reduce GHG emissions and enhance carbon sequestration in agricultural landscapes [19,20].

Conservation tillage and zero-tillage practices, for instance, are crucial in increasing soil carbon stocks, thereby contributing to the sequestration of atmospheric carbon dioxide. Studies by West and Post [21] have demonstrated that adopting conservation tillage can result in significant increases in soil organic carbon, which is essential for mitigating carbon dioxide emissions. Furthermore, Lal [22] emphasized the potential of soil carbon sequestration as a mutually beneficial strategy for both climate change mitigation and agricultural sustainability.

In addition, mixed cropping systems and the incorporation of cover crops are another sustainable practice with substantial benefits for GHG mitigation. These systems enhance soil structure, reduce erosion, and increase biodiversity, leading to more resilient agricultural ecosystems. Paustian et al. [23] highlighted that diversifying cropping systems can reduce nitrous oxide emissions and increase carbon storage in soils, contributing to the overall reduction of agriculture's GHG footprint.

Finally, agroforestry, or the integration of trees and shrubs into agricultural landscapes, is a sustainable practice that offers extensive carbon sequestration opportunities. In this context, Nair et al. [24] discussed the ways that agroforestry systems can capture carbon in both soil and biomass, providing a valuable tool for offsetting GHG emissions from agriculture. This multifunctionality of agroforestry systems not only addresses carbon sequestration, but also enhances biodiversity, improves water quality, and supports livelihood diversification.

The advent of 6G-IoT technologies creates unparalleled opportunities to further advance these sustainable practices. By enabling precise monitoring, data analytics, and automated control systems, 6G-IoT can optimize resource use, improve efficiency, and facilitate the adoption of practices that contribute to GHG emission mitigation. As a result, the integration of 6G-IoT in sustainable agriculture can transform traditional practices into intelligent, data-driven systems that are more responsive to environmental challenges and climate change mitigation goals.

## 2.2. Survey Scope

In contrast to the aforementioned studies, which primarily concentrate on traditional mitigation methods such as dietary modifications, management strategies, and breeding, the present study adopts a technologically oriented perspective and emphasizes the groundbreaking potential of 6G-IoT as a transforming technology, as presented in Table 1. Therefore, the primary contributions of this study are as follows:

- *Identifies and analyzes key GHG indicators*: It systematically identifies and evaluates the main sources and indicators of GHG emissions in agricultural practices, focusing on areas where emissions have increased as a result of intensive methods and increased chemical use. This provides a clear understanding of the problem areas that require mitigation.
- *Explores 6G-IoT's role in smart farming:* It delves into the evolving landscape of 6G-IoT technologies and their potential for reducing GHG emissions in smart farming practices. Furthermore, it provides an overview of technological advancements as well as real-world examples of how these technologies could be employed to reduce emissions.

- *Provides valuable insights:* By exploring the intersection of cutting-edge technologies and ecological sustainability, it identifies potential areas for future research and paves the way for further development in this critical field.

**Table 1.** Key contributions of related review studies and this work.

| Work | Year | Main Focus | GHG Emissions | IoT Integration | 6G Perspective |
| --- | --- | --- | --- | --- | --- |
| Nayak et al. [13] | 2015 | GHG emission mitigation | $CH_4$, $N_2O$ | ✗ | ✗ |
| Sejian et al. [12] | 2015 | GHG emission mitigation | $CH_4$, $N_2O$ | ✗ | ✗ |
| Koutsos and Menexes [15] | 2017 | Smart agriculture technologies | Not specific | ✗ | ✗ |
| Llonch et al. [14] | 2017 | Animal welfare and GHG emission mitigation | $CH_4$, $N_2O$ | ✗ | ✗ |
| Roy and George K. [16] | 2020 | Smart agriculture technologies | Full range | Partially | ✗ |
| Panchasara et al. [17] | 2021 | Smart agriculture technologies and GHG emission mitigation | Full range | ✓ | ✗ |
| Cheng et al. [18] | 2022 | Smart agriculture technologies | $CO_2$-eq | Partially | ✗ |
| Schwarz et al. [11] | 2022 | GHG emission mitigation | Not specific | ✗ | ✗ |
| Qian et al. [10] | 2023 | GHG emission mitigation | $CH_4$, $N_2O$ | ✗ | ✗ |
| **This Study** | 2024 | Smart agriculture technologies and GHG emission mitigation | Full range | ✓ | ✓ |

Therefore, this review emphasizes the revolutionary possibilities of 6G-IoT in smart farming, moving away from traditional mitigation strategies. Despite the identification of crucial emission indicators and the demonstration of practical applications, it sets the stage for further research and innovation, encouraging the agricultural industry to embrace technology for a more sustainable future.

## 3. Primary Contributors to Greenhouse Gas Emissions in Agriculture

Due to their significant contribution to global warming, GHG emissions are a primary topic of interest in the context of climate change. These emissions, consisting mainly of carbon dioxide ($CO_2$), methane, and nitrous oxide, are responsible for retaining energy in the upper atmosphere, hence contributing to the greenhouse effect. As illustrated in recent data from the U.S. Department of Agriculture in Figure 1, the agricultural sector alone accounts for a substantial 10% of these emissions, underscoring its impact on climate dynamics. This sector-specific contribution is vital to understand, given the potential for targeted mitigation strategies, as it can significantly influence the overall GHG landscape.

In the agricultural context, there are three main contributors to climate change, methane, nitrous oxide, and carbon dioxide, each derived from specific farming practices. In general terms, methane is mainly produced by livestock management, while the use of synthetic fertilizers and fossil fuels for agricultural machinery are the key contributors to nitrous oxide and carbon dioxide emissions, respectively.

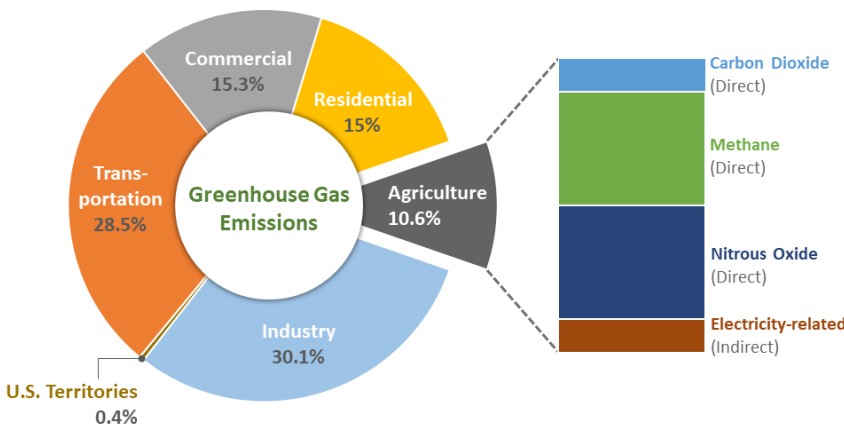

**Figure 1.** Global GHG emissions by economic sector. [*Source: U.S. Department of Agriculture, Economic Research Service, 2021*].

### 3.1. Methane Emissions

One of the key contributors to GHG emissions in agricultural applications is methane, a potent GHG gas that is mainly produced by ruminant livestock, including cattle, sheep, and goats, through a digestive process known as enteric fermentation. This natural digestive process involves methanogenic archaea in the stomachs of these animals, which aid in breaking down food, but also produce methane as a byproduct, which is then released into the atmosphere through belching. This process is inherent to the physiology of ruminants, making it a consistent source of methane emissions in livestock farming [25]. Additionally, rice cultivation is another notable contributor to methane emissions in agriculture. Since rice paddies are typically flooded for cultivation, they also create ideal conditions for an oxygen-depleted environment that favors the growth of methane-producing microorganisms, which thrive in the waterlogged soil and release methane into the atmosphere. As a staple crop globally, rice cultivation adds to the overall methane emissions from agriculture [26,27].

As a result, common practices applied to agriculture for methane mitigation firstly include dietary adjustments for livestock, such as the alteration of the ruminants' diet by incorporating fats, oils, and certain types of forages to redirect hydrogen in the rumen to pathways that do not produce methane [28,29]. Feed additives and supplements, such as nitrates and tannins, have also been explored to reduce methane emissions by altering the microbial processes in the digestive system of livestock. On the other hand, other studies have pursued advancements in breeding and genetics to develop livestock breeds that naturally produce less methane [30], with selective breeding programs aiming to identify genetic traits associated with lower methane emissions, offering a long-term approach to mitigation [31]. Finally, anaerobic manure digestion is a strategy proposed to mitigate methane emissions [32] by capturing methane from manure and converting it into bio-gas [33], which can be used as a renewable energy source. This not only reduces emissions, but also contributes to energy production, promoting sustainability in farming operations.

### 3.2. Nitrous Oxide Emissions

Another equally important GHG contributor in farming applications is nitrous oxide, which is a primary result of the use of synthetic and organic fertilizers. When fertilizers are applied to soils, especially in high-nitrogen-demanding crops, they undergo complex transformations, including nitrification, where ammonia-based fertilizers are converted into nitrate ($NO_3^-$) in aerobic conditions by soil microorganisms releasing nitrous oxide as a byproduct, and denitrification, where nitrate is reduced to gaseous forms, including nitrous oxide, under anaerobic conditions [34]. Similarly to methane, the magnitude of nitrous oxide emissions in agricultural operations is influenced by several factors, including the type and amount of fertilizer applied, soil properties, temperature, and moisture content.

As agriculture strives to meet the global demand for food, the use of synthetic fertilizers has increased significantly, consequently amplifying nitrous oxide emissions from agricultural soils. However, contrary to methane mitigation practices that focus on dietary adjustments, nitrous oxide mitigation strategies often revolve around optimizing fertilizer use and application methods [35], with precision agriculture techniques, such as site-specific nutrient management, aiming to tailor fertilizer application to crop needs, minimizing the excess nitrogen that can be converted into nitrous oxide [36,37]. Additionally, the use of nitrification inhibitors [38] and controlled-release fertilizers [39] can help reduce nitrous oxide emissions by slowing down the nitrification process in soils.

### 3.3. Carbon Dioxide Emissions

Finally, carbon dioxide emissions in agricultural practices are mainly associated with fossil fuels (e.g., gasoline and diesel) in farming machinery and equipment, whose combustion releases substantial amounts of carbon dioxide into the atmosphere [40]. Additionally, transportation activities related to the distribution of agricultural products to markets [41], deforestation for agricultural expansion [42], and the decay of organic matter in agricultural soils contribute to carbon dioxide emissions, further amplifying the sector's carbon footprint. Still, the significance of carbon emissions lies in their cumulative impact on climate change; while methane and nitrous oxide may have more immediate warming effects, carbon dioxide's extended atmospheric presence means that its contributions to global warming continue to accumulate over time; hence, its mitigation is deemed essential for reducing long-term climate change, as reducing its levels requires transitioning to cleaner and more sustainable farming practices, promoting the use of renewable energy sources for machinery and minimizing deforestation.

### 3.4. Understanding the Complexities of Agricultural GHG Emissions

To further understand the correlation between the aforementioned GHG indices and the primary sources of GHG emissions in agriculture, an overview table was created (Table 2) to offer a detailed examination of each source in conjunction with the corresponding indices, showcasing where and how these emissions are generated and how they can be measured and mitigated efficiently. It is evident that, although carbon dioxide is generally considered to be the main contributor to global warming due to its presence in many industrial and urban applications, in the agricultural context, carbon dioxide emissions are primarily derived from the combustion of agricultural residues and the use of agricultural machinery, which is probably the lowest percentage out of all the other gases caused by agricultural practices.

While carbon dioxide often dominates discussions about global warming, Table 2 reveals another perspective in the context of agriculture. Methane and nitrous oxide emerge as the two primary culprits, with methane originating from livestock farms, rice field flooding, and crop residue combustion. Nitrous oxide, on the other hand, is produced by livestock manure management, the use of synthetic fertilizers, and soil management practices. It is important to note, nevertheless, that carbon dioxide is a persistent greenhouse gas, which translates to prolonged atmospheric presence and a continuous warming impact. In contrast, methane and nitrous oxide have higher global warming potentials (GWPs) than carbon dioxide over shorter timescales, making them more potent in the short term. On this note, recent studies have shown that methane has a GWP of approximately 28–36 times greater than that of carbon dioxide over a 100-year period, while the respective GWP of nitrous oxide is around 265–298 times that of carbon dioxide over a 100-year timescale [43], having a much more immediate and intense warming effect despite its shorter atmospheric lifetime.

**Table 2.** Primary sources of GHG emissions associated with farming activities.

| GHG Emission Source | Description |
| --- | --- |
| Livestock | Methane ($CH_4$) emissions from the digestive process of ruminants |
| Manure Management | Emissions from stored or treated livestock manure |
| Synthetic Fertilizers | Nitrous oxide ($N_2O$) emissions from applying synthetic fertilizers |
| Rice Cultivation | Methane ($CH_4$) emissions from flooded rice fields |
| Soil Management | Nitrous oxide ($N_2O$) emissions from soil cultivation practices |
| Energy Use in Agriculture | Emissions from the use of fossil fuels in agricultural machinery |
| Crop Residues and Burning | Carbon dioxide ($CO_2$) and methane ($CH_4$) emissions from burning crop residues |

## 4. Greenhouse Gas Metrics in Agriculture

In the agricultural sector, the quantification and analysis of GHG emissions are crucial for developing appropriate mitigation strategies to combat climate change. GHG mitigation metrics in the agricultural sector, such as carbon footprint, emission intensity, GWP, carbon dioxide equivalents, methane emission rates, nitrous oxide emission factors, and energy use efficiency, serve as benchmarks for measuring the environmental impact of agricultural practices, posing essential tools for monitoring progress towards sustainable agriculture [44]. In Table 3, an overview of these key GHG mitigation metrics is offered, along with their respective agricultural field of application.

**Table 3.** Classification of key GHG mitigation metrics per smart farming application.

| GHG Metric | Description | Key Field of Application |
| --- | --- | --- |
| Carbon Footprint | Measuring total GHG emissions throughout farming activities and product lifecycles | Irrigation practices<br>Livestock management<br>Crop cultivation and harvesting<br>Farm-to-market transportation |
| Emission Intensity | Quantification of GHG emissions per unit of agricultural output | Wheat production per hectare<br>Meat production per dairy cow<br>Tomato yield per greenhouse area<br>Dairy and dairy product production |
| GWP | Evaluation of the warming effect of GHGs over a specified time compared with $CO_2$ | Land-use practices<br>Rice paddies emissions<br>Crop residue management<br>Synthetic fertilizer application |
| $CO_2$ Equivalents | Expressing GHG emissions in terms of $CO_2$ for easier comparison | Livestock enteric fermentation<br>Synthetic fertilizer application<br>Farm energy usage (fossil fuels)<br>Land-use-induced soil degradation |
| $CH_4$ Emission Rates | Tracking the rate at which methane is released | Rice paddies during flooding<br>Manure management systems<br>Enteric fermentation from cattle<br>Anaerobic digesters in livestock |
| $N_2O$ Emission Factor | Measuring emissions from synthetic and organic fertilizers | Livestock operations<br>Organic fertilizer application<br>Synthetic fertilizer application<br>Soil nitrification and denitrification |
| Energy Use Efficiency | Assessing the efficient use of energy in farming practices | Tractor operations<br>Irrigation pump systems<br>Renewable energy integration<br>Cold storage for perishable crops |

### 4.1. Carbon Footprint

More specifically, carbon footprint quantifies the total amount of GHGs produced directly and indirectly by farming activities or across a product's lifecycle, being a comprehensive measure that encompasses all GHG emissions associated with production, including carbon dioxide, methane, and nitrous oxide, from various sources such as fuel use, livestock, fertilizer application, and crop residue decomposition [45]. It also facilitates the comparison of different agricultural practices and production systems, allowing for the assessment of the environmental impact of various farming techniques, such as conventional agriculture versus organic farming [46], or the comparison of different livestock management practices. As a result, farmers can make informed decisions that prioritize environmentally sustainable practices by evaluating the environmental impact of their operations and implement changes to reduce emissions, opting for more efficient machinery, adopting precision agriculture techniques to optimize resource use [47], or implementing conservation practices such as no-till farming to sequester carbon in the soil [48].

### 4.2. GHG Emission Intensity

On the other hand, the emission intensity offers valuable insights into the efficiency of farming practices by measuring the amount of GHG emissions per unit of output (e.g., per kilogram of crop or meat produced), allowing for a comprehensive assessment of sustainability in the agricultural sector. By calculating the environmental efficiency of agricultural systems [49], this GHG mitigation metric allows for comparisons between different production methods and systems, helping farmers to identify opportunities for emission reduction and spot areas where GHG emissions can be reduced, while maintaining, or even increasing, agricultural productivity [50]. Additionally, emission intensity serves as an indicator of progress toward sustainability goals and can be applied to various aspects of agriculture, from assessing emission efficiency in crop production to livestock farming [51].

### 4.3. Global Warming Potential (GWP)

Another key GHG mitigation metric is the GWP that standardizes the assessment of the relative impact of different GHGs and serves as a crucial tool for evaluating the contributions of GHGs to climate change and guiding mitigation efforts within the agricultural sector. At its core, GWP quantifies the heat-trapping capability of GHGs in comparison with carbon dioxide, which is used as the baseline with a GWP value of 1, offering insights regarding other GHGs such as methane and nitrous oxide, as described previously. The GWP metric serves as a critical tool for setting emission reduction targets and policies by providing a standardized measure for comparing the impact of different GHGs, helping farmers to assess the relative significance of emissions from various sources and prioritize mitigation strategies accordingly [52].

By assigning GWP values to GHGs, GWP enables a more comprehensive understanding of the climate implications of different emission sources within the agricultural sector, highlighting the need for integrated and holistic approaches to GHG mitigation practices. Since different GHGs have varying GWP values, addressing emissions from multiple sources and gases can lead to more effective climate change mitigation [53]. For instance, reducing methane emissions from enteric fermentation in livestock [54] and simultaneously decreasing nitrous oxide emissions from synthetic fertilizer application [55] can have a more significant impact on GWP reduction than focusing on one gas alone.

### 4.4. Carbon Dioxide Equivalents

Carbon dioxide equivalents are used as a unified measure that accounts for the varying GWPs of different GHGs, playing a fundamental role in assessing and mitigating the environmental impact of agricultural activities by providing a standardized metric for comparing the total GHG emissions from various sources within agriculture. However, unlike the aforementioned GWP metric that measures the relative impact of individual GHGs compared with carbon dioxide, this metric aggregates the total emissions from all GHGs,

expressing them as if they were carbon dioxide emissions with equivalent GWP, hence simplifying the assessment of the overall climate impact of agricultural practices. Carbon dioxide equivalents help set reduction targets and track progress toward sustainability goals [56] and account for both direct (e.g., livestock or machinery emissions) [57] and indirect emissions (e.g., land-use changes, transportation, and energy use) [58,59].

*4.5. Methane and Nitrous Oxide Emission Rate*

Contrary to the emission intensity metric that is generally used for comparing the GHG efficiency of different farming systems or practices, methane emission rates and the nitrous oxide emission factor specifically represent the amount of the respective GHGs produced. More specifically, methane emission rates measure the amount of methane emissions produced per unit of time, while the nitrous oxide emission factor represents the amount of nitrous oxide emissions produced per unit of a specific activity or input in agriculture. Both of these metrics are especially important for GHG mitigation planning in agriculture, due to both methane's and nitrous oxide's significant contributions to the overall GHG emissions in the agricultural sector. However, the main difference is that methane emission rates are mainly focused on livestock, especially ruminants such as cows and sheep [54], as well as rice paddies [60], and measure the specific amount of methane emitted, while the nitrous oxide emission factor is proposed for fertilization practices [55], helping to identify areas where emissions can be reduced by optimizing fertilizer management.

*4.6. Energy Use Efficiency Index*

Finally, the energy use efficiency index is another key GHG mitigation metric that assesses GHG emissions related to energy consumption across various farming operations by measuring how efficiently energy is utilized in farming activities, with a focus on minimizing GHG emissions per unit of energy used [61]. Consequently, understanding and utilizing this metric can lead to the development of more efficient and environmentally friendly farming systems [62], having an impact on carbon footprint reduction, resource conservation, economic sustainability, as well as renewable energy and precision agriculture practices integration.

**5. The Role of 6G-IoT in Sustainable Farming**

The advent of IoT technologies has gradually transformed the landscape of smart agriculture, with each generation of mobile technology, 4G, 5G, and now 6G, playing an increasingly important role in this transformation. The transition from 4G to 6G communications reflects a narrative of technological evolution, with each phase contributing uniquely towards enhancing efficiency, productivity, and sustainability in agricultural practices.

*5.1. Evolution from 4G to 6G for Smart Agriculture*

The integration of 4G communications and IoT technologies marked a significant milestone in the evolution of smart agriculture, providing the foundation for improved connections across large agricultural landscapes. This combination facilitated the introduction of real-time data collection and remote monitoring capabilities, initiating a new era in farming that leveraged IoT devices for basic data analytics and machine-to-machine (M2M) communication [63]. The implementation of 4G-IoT in agriculture represented a significant change towards more data-driven decision-making processes. However, despite these advancements, the era of 4G-IoT encountered challenges, particularly regarding speed and latency, limiting the scalability of IoT applications and constraining real-time decision-making in complex agricultural environments.

The transition to 5G-IoT technology introduced a new level of connectivity, characterized by significantly higher data transmission speeds and reduced latency [64]. This advancement enabled the deployment of more complex applications that are critical to agriculture's future, including precision agriculture technologies, the utilization of un-

manned aerial vehicles (UAVs) for crop monitoring and spraying, as well as the application of real-time analytics, representing a significant increase in operational efficiency and resource optimization, fundamentally changing the way resources were managed. With 5G-IoT, sophisticated monitoring and control systems became feasible, directly affecting GHG emission mitigation by facilitating the precise management of water, fertilizers, and energy [65]. As a result, the advent of 5G-IoT technology not only improved the precision and efficiency of agricultural practices, but also contributed to the sector's sustainability goals by reducing the environmental footprint of farming operations.

The transition from 4G-IoT to 5G-IoT to 6G-IoT technologies is an ongoing progression of increasing capabilities, with each step ahead moving closer to achieving the full potential of IoT in agriculture. The advancements made with 5G-IoT, particularly in enhancing connectivity and data handling, have opened the way for a more integrated, efficient, and sustainable approach to farming practices. However, moving towards incorporating 6G-IoT technologies, the expectations are not merely improvements in speed and capacity, but a transformative shift towards highly autonomous, AI-driven agricultural ecosystems.

### 5.2. Introducing 6G-IoT: Technologies Transforming the Agricultural Landscape

Building on the capabilities of 5G, 6G-IoT technologies can bridge the gap between smart agriculture and sustainable farming practices by integrating IoT solutions that hold immense potential for mitigating GHG emissions, as [66] highlights. This approach offers a practical and sustainable solution by enabling real-time, data-driven monitoring of farm operations, leading to more efficient resource management and reduced emissions. Building on the foundation laid by 5G networks, which revolutionized connectivity and IoT device integration, 6G technology promises a transformative leap in communication systems [9]. However, the transition to 6G-IoT goes beyond simply increasing data rates and capacity. As illustrated in Figure 2, it serves as a catalyst for groundbreaking technologies with the potential to revolutionize agriculture and further improve GHG reduction.

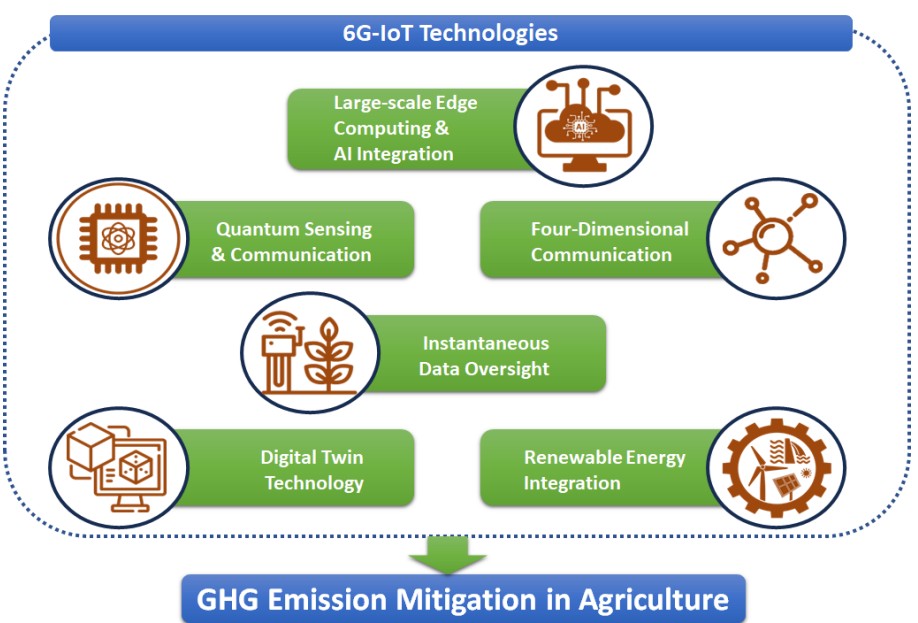

**Figure 2.** 6G-IoT technologies as the backbone for GHG emission mitigation in smart farming.

- The integration of *large-scale edge computing and artificial intelligence (AI)* will enable the analysis of sensor data in real time, allowing for more precise irrigation, optimized fertilizer use, and targeted pest control.

- *Quantum sensing* has the potential to revolutionize the field of agriculture by providing incredibly precise measurements of soil moisture, nutrient levels and plant health. This level of accuracy will enable targeted interventions and reduce unnecessary waste.
- Furthermore, *Quantum communication* will ensure the secure transmission of this highly sensitive data, protecting it from eavesdropping and manipulation.
- *Four-dimensional communication* will ensure extensive coverage and ultra-fast transmission of sensitive data, such as real-time crop health information from remote sensors, fostering efficient decision-making.
- Moreover, instantaneous data oversight will empower the predictive maintenance of agricultural equipment, minimizing downtime and ensuring optimal operation, while *digital twin technology* will enable the creation of virtual farms, allowing for crop yield simulations and risk mitigation strategies.
- Finally, *renewable energy integration* will power these technologies sustainably, minimizing agriculture's environmental footprint.

In the following sections, we delve deeper into the ways in which 6G-IoT technologies will open up new possibilities for advancements in future agriculture.

### 5.2.1. Hyperconnectivity and Ultra-Reliable Systems

Therefore, the advent of 6G-IoT will mark a critical advancement in smart agricultural technologies by introducing new standards in hyperconnectivity and reliability for farming and resource management systems, supporting massive amounts of data transfer with minimal latency, across a wide range of frequencies and with the capacity to connect to a vast number of devices simultaneously, thereby ensuring continuous, uninterrupted data flow, even in the most challenging and remote agricultural environments [67]. 6G-IoT will also facilitate the use of high-definition video, UAVs, and remote sensing technologies, offering detailed surveillance and monitoring of crops and livestock, enabling a multi-dimensional view of the farm, where multiple layers of data are integrated and synthesized to provide a comprehensive understanding of the agricultural ecosystem.

This ultra-high dimensionality, in a smart agricultural context, is related to the extensive deployment of sensors and devices across agricultural fields, capable of collecting real-time data on soil moisture, nutrient levels, crop health, and environmental conditions, which can later be combined to inform precise farming decisions. In addition, the ultra-high reliability of 6G-IoT systems is crucial for ensuring that the smart farming operations are consistently available, resilient, and fault-tolerant, allowing farmers and agricultural businesses to depend on the continuous operation of sensors, actuators, and autonomous vehicles, without fear of downtime or data loss. With 6G networks, this high data reliability is promised with the evolution of 5G ultra-reliable low-latency communications (URLLC) [68] that aim to provide even stricter latency and reliability requirements. Thus, it will be ensured that data transmission between devices and decision-making systems will not only be fast, but also consistent and dependable, thereby supporting the implementation of real-time control systems for automated machinery and allowing for precise operations.

### 5.2.2. Enhanced Data Accuracy and AI-Powered Decision-Making

In the context of smart agricultural applications, 6G-IoT will also significantly enhance various aspects of farming and resource management by improving both sensing accuracy and reliability, allowing wireless sensor networks to collect more detailed and accurate information, hence facilitating better decision-making practices and resource allocation strategies. Even in remote locations where connectivity has traditionally posed a challenge, 6G-IoT will allow farmers in isolated areas to optimize land use, manage resources more effectively, and enhance crop yield by facilitating advanced remote monitoring and diagnostics, as well as the use of augmented reality for enhanced management and training practices in agriculture. The introduction of 6G-IoT is expected to accelerate the adoption of advanced AI-based systems in agriculture, inherently changing the way GHG emission mitigation practices are approached. These systems will be able to leverage the vast

amount of data generated by IoT devices to autonomously perform complex tasks with unparalleled accuracy and efficiency. This includes disease detection, precision spraying, crop monitoring, and optimal planting predictions with minimum water consumption and chemical inputs, thereby directly reducing the carbon footprint of agricultural practices and, ultimately, enhancing operational precision and elevating the sustainability of farming operations.

Another notable innovation is the full integration of blockchain technology, which promises secure, transparent, and efficient transaction systems [69] that, when combined with the advanced capabilities of 6G-IoT, can address a variety of challenges such as data security, privacy, and trust in highly interconnected and autonomous systems. As a result, it can revolutionize supply chain management and traceability, ensuring data integrity from farm to table, thus supporting sustainable agricultural practices by promoting efficient resource usage and environmental responsibility [70].

### 5.2.3. Dynamic Renewable Energy Orchestration

The integration of 6G-IoT technologies into smart agricultural applications can also revolutionize the way renewable energy is orchestrated, making the management of energy resources (e.g., solar, wind, and biomass) more dynamic, sustainable, and efficient to meet the operational demands of agricultural activities [9,71]. More specifically, 6G-IoT enables the implementation of automated control systems for managing renewable energy resources in smart farming that can dynamically adjust energy distribution and storage based on real-time data and predictive analytics, ensuring that renewable energy is available when and where it is needed most, significantly reducing reliance on fossil fuels. By utilizing IoT devices and 6G connectivity, smart farming can leverage advanced predictive analytics and ML algorithms to forecast energy demands and renewable energy production by taking into account various factors, such as crop cycles, weather patterns, and historical energy usage data [72], to predict future energy needs and optimize the use of renewable energy accordingly. The case of solar-powered automated irrigation systems [73,74], for instance, exemplifies the way that the integration of 6G-IoT technologies can lead to substantial energy savings and a lower GHG footprint, not only aligning with sustainable farming practices, but also showcasing the critical role of smart technologies in achieving environmental targets. Such predictive approaches ensure that renewable energy resources are utilized efficiently, reducing waste and enhancing the sustainability of farming operations.

Furthermore, smart farming operations equipped with 6G-IoT technologies can also participate in energy trading and grid integration more effectively by not only optimizing their own energy consumption, but also supplying excess renewable energy back to the grid, thanks to 6G's real-time data and reliable connectivity facilitation.

## 6. Harnessing 6G-IoT's Precision Power: Tailored Solutions for GHG Emissions

In Section 4, various GHG metrics that are critical for assessing and managing emissions in agriculture have been introduced. However, while these metrics serve as foundational tools for understanding and quantifying GHG emissions, the transformative potential of 6G-IoT technologies in impacting these metrics justifies further exploration. For this purpose, this section showcases how 6G-IoT can revolutionize various aspects of agriculture, driving significant reductions in GHG emissions while optimizing resource use and boosting productivity. From enhancing soil carbon sequestration and reducing methane emissions from livestock to integrating renewable energy sources and revamping precision agriculture, 6G-IoT paves the way for a future of sustainable farming practices that contribute to a healthier planet and a more secure food system [6]. A comprehensive outlook of the way 6G-IoT technology can be integrated into smart farming applications for GHG emission mitigation, along with the respective benefits offered, is depicted in Figure 3.

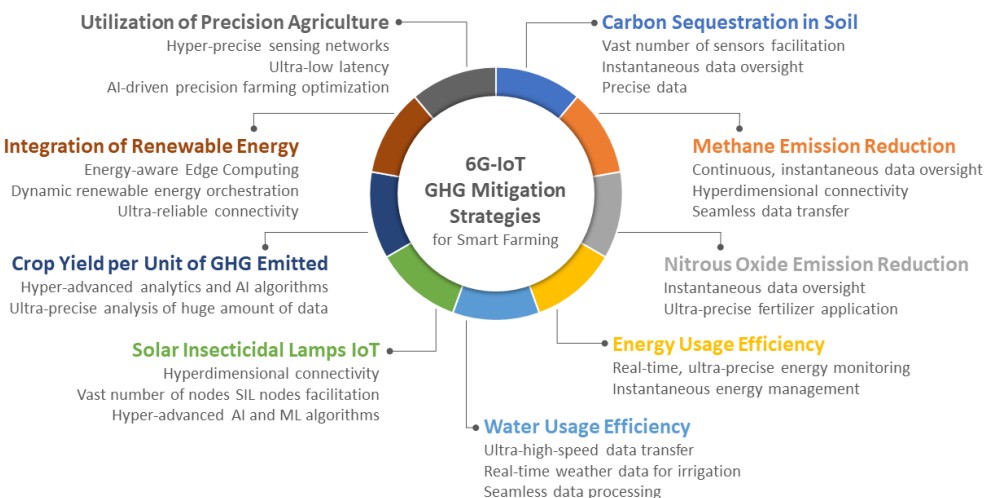

**Figure 3.** 6G-IoT integration and its expected benefits in common GHG emission mitigation strategies in smart agriculture.

### 6.1. Carbon Sequestration in Soil

Carbon sequestration in soil is a vital strategy in mitigating GHG emissions and combating climate change [22]. By enhancing carbon storage in agricultural soils, the concentration of atmospheric carbon dioxide can be effectively reduced, while the integration of 6G-IoT technologies into smart farming practices offers innovative solutions to enhance carbon sequestration.

6G-IoT technology's facilitation of the connection of a vast number of sensors to a single tower allows for continuous, instantaneous monitoring of farming activities and their associated emissions, providing precise and continuous data that can be later used to identify carbon-intensive practices and transition to more sustainable alternatives. As a result, farmers can minimize carbon emissions during soil cultivation through reducing the use of heavy machinery and opting for no-till farming. The data-driven architecture of the 6G-IoT-enabled smart farming supports access to real-time climate data and emissions information even in remote areas, allowing farmers to make informed decisions that prioritize crops or practices with lower GWPs, such as perennial crops that sequester carbon for longer periods, as well as to calculate and track carbon dioxide equivalents in real time. As a result, farmers can choose practices that result in net carbon sequestration, such as afforestation or reforestation of marginal lands. Finally, since 6G-IoT also enables precise measurements thanks to its quantum sensing technology [9], it inherently allows farmers to optimize resource usage when monitoring GHG emission intensity on a per-unit basis (e.g., per crop yield or area), and minimize emissions while maintaining productivity.

### 6.2. Methane Emission Reduction in Livestock

Reducing methane emissions from livestock is a critical aspect of mitigating GHG emissions in agriculture, since livestock, especially ruminants, are significant contributors to methane emissions due to enteric fermentation during digestion and manure management [75]. However, the integration of 6G-IoT technologies into livestock management presents new innovative solutions for methane reduction, hence enabling more targeted and effective methane reduction strategies.

In livestock management, the continuous and instantaneous monitoring characteristic of the 6G-IoT technology [64] allows farmers to repeatedly measure methane emission rates from individual animals, helping them identify high emitters and implement dietary adjustments, as well as to adapt to livestock management practices based on expected or unexpected weather conditions. After all, minimizing stress on livestock can reduce emissions and lower the GWP. Furthermore, the provision of additional data on the entire

lifecycle of the livestock is also feasible, thanks to the newly introduced four-dimensional communication characteristic of the 6G-IoT technology, which expands geographical coverage beyond the current boundaries of 5G networks, thus allowing farmers to make more informed decisions about breeding, nutrition, and management practices. Finally, quantum sensing technology and the integration of edge computing with AI can guide farmers in improving production efficiency and create automated procedures [76], ultimately leading to reduction in emissions per unit of output, and energy consumption and methane emissions associated with energy production, respectively.

### 6.3. Nitrous Oxide Emission Reduction

As in methane emission mitigation, 6G-IoT can also aid in the reduction of nitrous oxide emissions by integrating new and innovative solutions [6], making the already existing mitigation practices more effective, while also promoting both environmental sustainability and agricultural productivity.

As already mentioned in the previous subsection, the quantum sensing technology presented by the emergence of the 6G-IoT integration in smart farming applications ultimately leads to more precise measurements due to the enhanced sensitivity offered compared with conventional field sensors. As a result, farmers will be able to use precise data of soil conditions and nutrient levels to both optimize fertilizer application and fine-tune their application rates, reducing excess nitrogen that can lead to nitrous oxide emissions and minimizing the overall emission intensity [77]. Furthermore, variable rate technology (VRT), also introduced in by 6G-IoT, can help in enabling the precise application of fertilizers, matching them to the specific needs of each field or even subfield [78]. On the other hand, the real-time monitoring characteristic of 6G-IoT can aid in carbon footprint reduction, GWP mitigation, and energy use optimization as well, by optimizing fertilizer application. Consequently, farmers will be able to minimize nitrous oxide emissions, adapt fertilizer application schedules to to match optimal conditions for nutrient uptake by crops, and reduce the likelihood of nitrous oxide emissions associated with nitrogen loss.

### 6.4. Enhancing Energy and Water Usage Efficiency

Efficient energy and water usage in agriculture is a key element of GHG mitigation strategies, since the agricultural sector not only relies on various energy sources, including fossil fuels for machinery and electricity for various operations, but also accounts for a significant portion of the global water consumption. Reducing energy and water consumption not only lowers GHG emissions, but also contributes to sustainable and cost-effective farming practices [79]. On that note, the integration of 6G-IoT technologies offers innovative solutions for optimizing energy and water use in agriculture, allowing for more targeted and effective reduction strategies.

Aside from continuous, real-time monitoring of crops and farms, 6G-IoT also offers real-time data on energy (machinery usage and electricity consumption) and water consumption across the farm by continuously monitoring usage patterns and identifying areas of high consumption, which can then be used to lower the system energy input and to schedule energy- and water-intensive operations during off-peak hours. In addition, the provision of precise data related to energy use efficiency can help farmers to assess their system's energy use efficiency in relation to their production, reducing excessive emission intensity, and track its energy use at various stages of production to identify areas where energy use can be optimized.

Finally, 6G-IoT can also enhance energy efficiency by integrating renewable energy sources (e.g., solar panels and wind turbines) into agricultural operations [80], as well as exploiting smart grids and AI algorithms to manage the generation and distribution of renewable energy on farms, promoting sustainability and reducing reliance on fossil fuels. The integration of AI algorithms can also play a pivotal role on the development of precise smart irrigation systems [81] that limit excess water consumption and follow watering patterns based on plant needs and weather data. The digital twin technology could also aid

in that direction by offering representative simulated examples [9] of energy usage patterns on specific farming operations without the need for real-world deployment during testing, hence avoiding excess energy consumption.

### 6.5. Enhancing Solar Insecticidal Lamps IoT

Solar Insecticidal Lamps IoT (SIL-IoTs) is a sustainable smart farming approach that traditionally relies on 4G, Wi-Fi, and ZigBee communications for real-time data transmission, thereby being critical in monitoring and managing pest activity across agricultural fields. However, the advent of 6G-IoT technologies promises revolutionary enhancements, drastically improving the pre-existing real-time data processing, expanding connectivity, and, ultimately, increasing the precision of pest control operations.

Typically, SIL-IoTs combine solar energy with IoT technologies to effectively manage pest populations, leveraging pests' phototactic responses instead of regular pesticides [82], thereby already contributing to GHG mitigation. However, the integration of 6G communications offers a new age of URLLC and massive Machine-Type Communications (mMTC), thereby facilitating instantaneous data processing and ultra-long-range communications. As a result, dynamic adjustments of SIL-IoT operations in response to real-time insect detection, as well as the deployment and coordination of a vast number of SIL nodes across extensive agricultural fields can be easily supported [83], ensuring that SIL-IoTs are only activated during peak pest activities, thus conserving energy and optimizing pest eradication efficiency. In addition, advanced AI and ML algorithms can also be implemented for analyzing pest phototactic rhythms and environmental conditions, allowing for the dynamic modification of SIL operational parameters, such as light intensity, spectrum, and pattern, to tailor pest eradication strategies effectively while minimizing energy consumption [84].

Finally, 6G-IoT's advanced positioning and communication capabilities can help overcome limitations of traditional SIL-IoTs by providing accurate pest detection and localization even under challenging conditions, such as asymmetric communication links [85]. The application of such technologies ensures that pest control efforts are concentrated in areas where they are most needed, thereby optimizing resource utilization and enhancing the efficiency of pest management interventions.

### 6.6. Optimizing Crop Yield per Unit of GHG Emitted

Enhancing crop yield while minimizing GHG emissions is a critical goal in modern agriculture. One effective approach includes measuring the Crop Yield per Unit of GHG Emitted (CYE), a GHG mitigation metric that quantifies the productivity of agricultural practices relative to their emissions [86]. However, the integration of 6G-IoT offers advanced tools to improve CYE and contribute to sustainable farming, encompassing data-driven decision-making and automation.

As in previous mitigation strategies, the provision of continuous and instantaneous data on emissions from various farming activities, such as fuel use, fertilizer application, and livestock management, from 6G-IoT allows for carbon footprint reduction and GWP mitigation by increasing crop yields while minimizing emissions. In addition, the integration of advanced AI- and ML-based algorithms allows for automated and optimized farming practices with minimum energy and water consumption, leading to resource usage optimization, wastage reduction, and emission intensity minimization [5].

On the other hand, key IoT smart farming applications, such as precision agriculture, automated farming processes, and crop diversification, will also be enhanced with the integration of 6G-IoT communications. More specifically, high-resolution sensors and real-time data streams allow for precise resource management, while 6G-IoT-enabled ML-based operations encompass extensive dataset analysis, data-driven decision-making, and automation [6] to identify the most efficient farming practices for maximizing CYE. Finally, since 6G-IoT technology provides access to market data and climate forecasts, farmers can use this information to diversify crops and adapt to changing conditions, reducing the

vulnerability of agricultural systems to specific GHG emissions, such as those associated with monoculture practices.

### 6.7. Integration of Renewable Energy in Farming Operations

The integration of renewable energy sources into farm operations is a crucial step in mitigating GHG emissions and promoting sustainability in agriculture, not only by reducing carbon emissions, but also by contributing to energy self-sufficiency [87]. The merge of renewable energy sources with 6G-IoT technology allows the implementation of renewable energy solutions to become more efficient and effective, representing a significant step towards sustainable and low-carbon agriculture [88].

More specifically, by utilizing continuous, real-time data and smart solutions, farmers can optimize energy use, reduce GHG emissions, and lower their carbon footprint [64]. For instance, collecting and analyzing real-time data associated with energy consumption and carbon emissions from various sources, including machinery and heating and cooling systems, future areas of improvement can be identified, resulting in lowering the carbon footprint of the respective operations. The integration of smart grids along with the quantum sensing technology and terahertz (THz) communications extends the system's energy management, allowing for energy-efficient devices that can be securely controlled from remote areas without utilizing massive amounts of energy [9].

On that note, the revised form of precision agriculture supported by 6G-IoT also extends to energy management, allowing farmers to use real-time data on weather conditions, crop growth, and energy demand to optimize the use of renewable energy. For instance, excess energy generated during sunny days can be stored or redirected to power irrigation systems or farm equipment. This precision energy management reduces the reliance on fossil fuels and lowers GHG emissions.

### 6.8. Utilization of Precision Agriculture Technologies

The utilization of precision agriculture technologies in farming practices has the potential to significantly mitigate GHG emissions while simultaneously improving agricultural productivity [36]. However, when coupled with the capabilities of 6G-IoT technology, precision agriculture becomes even more effective in reducing GHG emissions by optimizing resource use, improving energy efficiency, and minimizing the carbon footprint. As a result, 6G-IoT-enabled precision agriculture aligns with global efforts to combat climate change and GHG mitigation in farming applications.

Generally, precision agriculture minimizes the carbon footprint by optimizing resource usage, including water, fertilizers, and pesticides. In this context, the integration of 6G-IoT technology plays a crucial role by providing continuous, real-time, and secure data on soil conditions, weather patterns, and crop health, allowing farmers or automated (ML-based) smart farming systems to make informed decisions regarding irrigation, nutrient application, and pest control, resulting in reduced GHG emissions associated with resource overuse [5,89]. Additionally, the facilitation of data-driven decision-making by collecting information from various sensors and devices deployed in the field aids in the optimization of crop yields while minimizing resource inputs, ultimately reducing emission intensity decreases, since the same or higher agricultural output is achieved with fewer emissions.

6G-IoT-enabled precision agriculture also contributes to the reduction of GWP by employing precise smart irrigation and energy management systems, optimizing land use, and reducing deforestation. Real-time data on crop growth and soil health enable farmers to maximize yield per unit of land, reducing the need for land expansion, while the use of precision agriculture minimizes the use of synthetic fertilizers, which are a significant source of nitrous oxide emissions with high GWP. Finally, 6G-IoT enhances precision agriculture by employing high-resolution sensors and super-high data transmission, allowing for ultra-remote monitoring and control of energy-intensive processes, as well as enhanced remote nutrient and disease assessment [9].

**7. Conclusions: Key Insights and Interdisciplinary Implications towards Sustainability**

While smart agriculture and sustainable agriculture may initially appear as distinct concepts, the adoption of 6G-IoT technologies in precision agriculture signals a paradigm shift in smart agriculture's strategy for reducing GHG emissions, hence leading to more sustainable practices and environmentally friendly solutions. This evolution aligns with the collaborative efforts of global organizations, including the Organization for Economic Co-operation and Development (OECD) and the Food and Agriculture Organization of the United Nations (FAO), which consistently analyze and propose policies meant for promoting fair and sustainable agricultural growth. As stated in their joint OECD–FAO Agricultural Outlook reports [90,91], these organizations underscore the importance of enhancing agricultural productivity through state-of-the-art research, sustainable practices, and supportive trade policies.

However, since the demand for agricultural products is constantly rising due to global dynamics such as population growth, urbanization, and dietary preferences, new and innovative approaches must be developed. This is where 6G-IoT-enabled precision agriculture comes into the picture, which, thanks to its facilitation of an unmatched level of data collection and analysis, in addition to the deployment of extensive sensor networks and quantum communications, has the ability to completely transform contemporary precision agriculture by enabling instantaneous and precise decision-making [92,93]. This new fusion of precision agriculture practices can be considered the backbone of GHG emission mitigation practices, leading to greener and more sustainable farming applications.

Additional advancements that are expected with integrating 6G-IoT in smart farming include state-of-the-art edge computing and AI-based models for enhancing disease detection, treatment applications, and overall plant health management with greater precision and efficiency, a lot faster. These innovations, supported by quantum and four-dimensional communications provided by 6G networks, will not only improve communication efficiency and facilitate advanced computational models, but will also significantly reduce the carbon footprint associated with traditional farming practices by supporting the deployment of wide-area environmental monitoring systems [94,95]. Furthermore, the integration of 6G-IoT's high-speed and low-latency communication capabilities will also enhance the functionality and application of the digital twin technology, offering resource efficiency, enhanced decision-making with AI [96], as well as climate adaptation and reduced physical trials and errors, fostering a more sustainable agricultural model globally, minimizing the environmental impact of farming practices.

*7.1. Translating Science into Practice: Enhancing 6G-IoT Adoption in Agriculture*

While 6G-IoT holds immense potential to revolutionize sustainable agriculture, achieving widespread adoption remains a critical issue [97]. This requires a multidisciplinary approach that emphasizes knowledge sharing, active participation of Agricultural Knowledge and Innovation Systems (AKIS) [98], and engagement with change agents and farm advisors, with the objective of translating scientific advancements into actionable strategies for all stakeholders. To bridge the existing gap between technological potential and farmer adoption, an organized process involving AKIS is crucial. Such systems can serve as the backbone for advancing the understanding and implementation of 6G-IoT technologies by integrating research, education, and extension services. With a focus on usability and practical advantages, this approach ensures that innovations are not only produced but also efficiently distributed across the agricultural community. Moreover, the involvement of change agents is paramount in driving adoption. Agricultural extension officers, industry leaders, and innovative farmers play pivotal roles as catalysts for adoption. They could leverage their influence and networks to promote 6G-IoT technologies, demonstrating their positive impact on farm efficiency, productivity, and sustainability [99].

However, to optimize farmer adoption and, subsequently, reduce GHG levels, it is also essential to incorporate the insights and recommendations of farm advisory services [100]. Personalized guidance that considers regional contexts, challenges, and opportunities

may significantly enhance the relevance and appeal of 6G-IoT solutions, thereby helping farmers navigate the decision-making process, address concerns, and highlight the alignment of 6G-IoT with their specific needs and objectives. In conclusion, the effective implementation of 6G-IoT advancements in real-world applications necessitates a cohesive approach that involves AKIS, the engagement of influential change agents, and the availability of contextual and practical farm advisory services. Addressing these aspects will not only facilitate the adoption of 6G-IoT technologies, but also maximize their impact on the agricultural industry.

### 7.2. Challenges in 6G-IoT Adoption

Despite the significant advancements mentioned, integrating 6G-IoT technologies into smart farming applications will present a number of challenges. The primary barriers on this path are technological and infrastructural. However, there are other significant challenges to overcome:

- *Data management and integration:* Diverse data sources, from IoT sensors and satellites to weather stations, often lack compatibility, presenting challenges in seamless integration and interoperability. The lack of consistency in the data complicates the decision-making process and decreases the effectiveness of smart farming solutions.
- *Energy demands:* Powering 6G networks and the 6G-IoT infrastructure, especially in remote areas, requires a significant amount of energy. Finding sustainable and energy-efficient solutions becomes increasingly important, both financially and environmentally.
- *Regulatory compliance:* Ensuring compliance with evolving regulatory standards for data privacy, environmental regulations, and cybersecurity adds a new layer of complexity. Meeting these standards is critical, as noncompliance might compromise farming operations' efficiency.
- *Environmental impact:* It is critical to ensure that the proposed solutions do not worsen the problems they are meant to address. Thus, environmental impact reduction becomes an important consideration during the deployment and maintenance phases.
- *Human dimension:* Addressing skepticism and building trust among farmers towards 6G-IoT technologies is crucial for adoption, as misconceptions regarding the complexity, cost, or applicability of 6G-IoT solutions can hinder their use. Thus, simplifying the technology's benefits and proving its reliability are key.
- *Socio-economic and cultural considerations:* Tailoring 6G-IoT solutions to fit diverse socio-economic statuses and cultural practices of farmers ensures wider acceptance and usability. Such strategies must bridge technology gaps while respecting traditional agricultural knowledge.

While 6G-IoT holds immense potential to revolutionize agriculture and unlock sustainable, efficient practices, it is also critical to recognize their inherent technological and infrastructural complexity. For instance, although 6G-IoT offers unparalleled connectivity and data acquisition capabilities, it also amplifies existing challenges related to the interoperability and integration of diverse data sources.

This duality underscores the critical need for developing robust data integration frameworks as a foundational step in realizing the benefits of 6G-IoT in smart farming. For this purpose, achieving this vision demands collaborative efforts, strategic planning, and sustained investments in technology and infrastructure development. By overcoming these complications, 6G-IoT-enabled precision agriculture can become a pillar for mitigating greenhouse gas emissions in smart farming, leading to a more sustainable future for our planet and its food production.

### 7.3. Closing Remarks and Future Work

This review paper serves as a comprehensive analysis of key GHG indices that significantly contribute to emissions in the field of smart farming, as well as basic GHG mitigation metrics often taken into consideration during mitigation planning in the agricultural field, in order to provide a framework for measuring the impact of each GHG emission source

and guide the implementation of new GHG mitigation strategies. The integration of 6G-IoT technologies can further refine these strategies through real-time monitoring, data analytics, renewable energy integration, and automated control systems, paving the way for 6G-IoT-enabled precision agriculture and smarter resource management. As agriculture moves toward sustainability, the use of these indices, coupled with technological advancements, will be instrumental in achieving significant reductions in GHG emissions.

Finally, as a future direction, the development of an optimization function will be completed, which will not only aid in mitigating GHG emissions in smart agriculture, but also align with financial technology (FinTech) principles [101], aiming to integrate economic and environmental sustainability, providing a holistic solution to the challenges faced in modern smart agriculture.

**Author Contributions:** Conceptualization, S.P. and D.N.S.; investigation, S.P. and D.N.S.; writing—original draft preparation, S.P. and D.N.S.; writing—review and editing, S.P. and D.N.S.; supervision, D.N.S., P.S., G.K. and C.S. All authors have read and agreed to the published version of the manuscript.

**Funding:** This research received no external funding.

**Conflicts of Interest:** The authors declare no conflict of interest.

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
