# Peer review of "Smart Agriculture and Greenhouse Gas Emission Mitigation: A 6G-IoT Perspective"

_electronics, doi:10.3390/electronics13081480_

Round 1

Reviewer 1 Report

Comments and Suggestions for Authors

The manuscript well described the Greenhouse Gas Emission and the role of 6G-IoT. The analysis on  Harnessing 6G-IoT’s Precision Power is also expressed clearly and useful. It will benefit to readers who would like to learn knowledge for Greenhouse Gas issues and the advanced 6G-IoT. General speaking, the manuscript is ok. However,it is hard to judge whether or not this manuscript is suitable to published in Electronics.

1. The manuscript is mainly organized from the perspective of the concept of 6G-IoT and its advantages. The author did not present any of the technology of the 6G or IoT. 

2. For me, it is really hard to build direction linkages between the 6G-IoT with Greenhouse Gas Emission. 6G-IoT is able to support and facilitate better smart agriculture application. Could it directily mitigate the Greenhouse Gas Emission. I hold a skeptical attitude. It seems its function was a bit over emphasized.

The next is minor issues could be addressed easily.

3.  In section 4, the author introduced "Greenhouse Gas Metrics in Agriculture", however, there is no future introduction on how to use 6G-IoT to impact the Greenhouse Gas Metrics in the following sectors. 

4. The title is "Smart Agriculture and Greenhouse Gas Emission Mitigation: A 6G-IoT Perspective". However, no smart agriculture was futhure explained. The author employed Sustainable Farming or Sustainable Agriculture rather than Smart Agriculture.

Reviewer 2 Report

Comments and Suggestions for Authors

1. The title is about smart agricultural, however, there is no any description about it in introduction, please add them.

2. 6G-IoT is new technology, what is the difference with previous technologies, please add them in introduction.

3. What is the 6G-IoT, why choose this technology in this paper?

4. 2. RelatedWork and Survey Scope is a little simple, please improve it.

5. What is the relationship between the smart agricultural and the GHG, they seem that no any relevance. Please address them.

6. What is the smart theologies presented in this paper, they are mostly about the GHG and sustainability?

7. The conclusions and abstracts should be improved with more highlights.

8. The references in this review are not enough, many applications could be added for example.

Reviewer 3 Report

Comments and Suggestions for Authors

The article explores the impact of 6G-IoT technologies on sustainable agriculture, focusing on various key areas. Initially, it discusses the role of IoT and 6G in optimizing agricultural practices to reduce greenhouse gas emissions. It highlights the potential of edge computing, artificial intelligence, and quantum sensing technology in improving irrigation, fertilizer usage, and pest control.

Subsequently, it addresses the hyper-connectivity and reliability of 6G-IoT systems, enabling detailed monitoring of crops and livestock, as well as optimized resource management and emissions reduction. The importance of data precision and AI-based decision-making in enhancing agricultural productivity and sustainability is emphasized.

The text also delves into the effectiveness of 6G-IoT technologies in improving data accuracy and optimizing agricultural decisions. It highlights the benefits of integrating blockchain technology and energy forecasting for efficient resource management and increased supply chain traceability.

Moreover, it discusses the dynamic orchestration of renewable energies in agriculture and the potential of 6G-IoT technologies in reducing methane and nitrous oxide emissions associated with livestock activities and fertilizer management. The importance of efficient energy and water management in optimizing agricultural practices and reducing emissions is stressed.

Lastly, it highlights the key role of precision agriculture enabled by 6G-IoT technologies in improving crop yields while simultaneously reducing greenhouse gas emissions. It discusses challenges in adopting 6G-IoT technologies, such as data management, energy requirements, and regulatory compliance, but also underscores the revolutionary potential of these technologies in sustainable agriculture.

Personally, I found the article very interesting.

Reviewer 4 Report

Comments and Suggestions for Authors

The authors included an extensive literature review using mostly contemporary scholarship. This is the primary strength of the manuscript. 

Authors provided no less than 20 times the extent and specificity of 6IOT advantages to farmers. 

The large detail that is missing is 'how' the innovation will be diffused, knowledge transferred, or agricultural knowledge and innovation systems (AKIS) utilized to promote/encourage farmer adoption? The authors do, without specifically stating it, how their science informs practice. However, the authors do not provide how the science will be translated to stakeholders, thus creating outcomes and subsequent impacts. There was no mention of change agents or farm adivsory services. This missing elements inform us of the new science that was developed and available but not the recommendations or implications of including change agents or opinion leaders in farmer' innovation-decision process to optimize farmer adoption and consequently reducing GHG levels. 

The scholarship is timely, provides more easily scientific replications for scientists. However, the missing void of translating science to practice adoption to improve outcomes and impacts is a glaring hole in this scholarship. Page 15 provided four challenges to overcome. One large omission is the human dimension or the social and behavioral science challenge the manuscript does not include. Science is only science until we get stakeholders/actors to use the science, and then, only then might we produce outcomes from our science. 

Round 2

Reviewer 1 Report

Comments and Suggestions for Authors

The author has addressed the previous comments properly. However, there are still some issues needs to be clarified. Please see the following comments.

1. Regarding the title, why didn't the author go directly to Smart Agriculture or sustainalbe agriculture from the prospective of 6G-IoT. I am not able to get this point why the author chose the GHG Emissions. As I said in my previous review, There are no direct connections between GHG Emission and 6G. At least for me, I hold this standpoint.

2. If the author insist in the same title for this manuscript. I still have several suggestions.

2.1 What is the relationship between the GHG and sustainalbe agriculture? Should this be elaborated in the section "2. Related Work and Survey Scope"?

2.2 In section "3. Primary Contributors to Greenhouse Gas Emissions in Agriculture", the author introduced the GHG in agriculture. However, in the following section 5 and 6, the author did introduce the The Role of 6G-IoT in Greenhouse Gas Emissions. It goes to the sustainable agriculture. The scope and main study object is unclear. How about change the title of Section 6 to " Harnessing 6G-IoT’s Precision Power: Tailored Solutions for Greenhouse Gas Emissions"?

2.3 In section "4. Greenhouse Gas Metrics in Agriculture", I doubt the necessity to have this section. The reason is that no Greenhouse Gas Metrics are discussed in the following section. For example, how to use the 6G-IoT to reduce or impact the Greenhouse Gas Metrics. 

2.4 In line 698-701, "Data management and integration" is not the challenges in 6G-IoT adoption. On the contrary, 6G-IoT adoption will produce the problem, like Data management and integration.

Since I still have the aforementioned doubts. I recommend a mojar revision again. And would be happy to see the author's argument.

Reviewer 2 Report

Comments and Suggestions for Authors

The paper was revised well.

Author Response

We would like to thank the reviewer for their positive assessment of our previous revisions.

Reviewer 4 Report

Comments and Suggestions for Authors

The Conclusions sections were greatly improved with subsections 7.1 and 7.2. I believe these inclusions that enhanced theory to practice warrants the publication of the manuscript. 

Author Response

(The authors gave the same response as above.)

Round 3

Reviewer 1 Report

Comments and Suggestions for Authors

I don't have additional comments.

Author Response

(The authors gave the same response as above.)
